# Combined BSA-Seq and RNA-Seq Reveal Genes Associated with the Visual Stay-Green of Maize (*Zea mays* L.)

**DOI:** 10.3390/ijms242417617

**Published:** 2023-12-18

**Authors:** Ran Zheng, Min Deng, Dan Lv, Bo Tong, Yuqing Liu, Hongbing Luo

**Affiliations:** 1College of Agronomy, Hunan Agricultural University, Changsha 410128, China; zran@stu.hunau.edu.cn (R.Z.); 2402277692@stu.hunau.edu.cn (B.T.);; 2Maize Engineering Technology Research Center of Hunan Province, Changsha 410128, China

**Keywords:** maize, visual stay-green, BSA, RNA-seq, genes

## Abstract

Maize has become one of the most widely grown grains in the world, and the stay-green mutant allows these plants to maintain their green leaves and photosynthetic potential for longer following anthesis than in non-mutated plants. As a result, stay-green plants have a higher production rate than non-stay-green varieties due to their prolonged grain-filling period. In this study, the candidate genes related to the visual stay-green at the maturation stage of maize were investigated. The F_2_ population was derived from the T01 (stay-green) and the Xin3 (non-stay-green) cross. Two bulked segregant analysis pools were constructed. According to the method of combining ED (Euclidean distance), Ridit (relative to an identified distribution unit), SmoothG, and SNP algorithms, a region containing 778 genes on chromosome 9 was recognized as the candidate region associated with the visual stay-green in maize. A total of eight modules were identified using WGCNA (weighted correlation network analysis), of which green, brown, pink, and salmon modules were significantly correlated with visual stay-green. BSA, combined with the annotation function, discovered 7 potential candidate genes, while WGCNA discovered 11 stay-green potential candidate genes. The candidate range was further reduced due through association analysis of BSA-seq and RNA-seq. We identified *Zm00001eb378880*, *Zm00001eb383680,* and *Zm00001eb384100* to be the most likely candidate genes. Our results provide valuable insights into this new germplasm resource with reference to increasing the yield for maize.

## 1. Introduction

Maize (*Zea mays* L.) is one of the world’s three major food crops. As well as being an important resource for food and industrial output, it plays a visible role in guaranteeing national food security and fulfilling market demand. The improvement in grain yield is a longstanding and important breeding target for maize [1]. Some mutants, known as “stay-green mutants”, retain a typical green phenotype even after leaf senescence. The stay-green mutants’ capacity to maintain photosynthesis and collect more photosynthetic products during the later phases of growth has led to speculation that stay-green-related features may be useful in increasing crop yields further [2,3,4]. A better understanding of the genetic mechanisms underlying stay-green maize may lead to applications that increase maize yield.

Several studies indicate that the stay-green trait is a complex quantitative trait in maize [5,6,7,8,9]. Over the previous decade, studies on the stay-green trait of maize have mostly focused on the QTL (quantitative trait locus) placement of stay-green-related traits. More than 200 QTLs are associated with the stay-green trait in maize. The QTLs for stay-green-related traits are distributed on chromosomes 1–10. For example, Fang et al. [10] conducted a meta-analysis on stay-green QTLs identified in the studies of Liu et al. [11], Wang et al. [12], and Zheng et al. [13], and five consensus QTL intervals (bin1.04, bin1.07, bin4.01, bin5.03, bin9.03) for the maize stay-green trait were identified. Yang et al. [14] used an F_3:4_ population to map the QTLs for maize stay-green traits and identified 23 QTLs distributed on nine chromosomes. Kante et al. [15] identified the QTLs for the stay-green trait in bins 1.04–1.09, 5.02, and 10.04–10.06 and confirmed that bnlg1556 on chromosome 1 is closely linked to stay-green traits. However, there are notable differences in the numbers of QTLs detected in different experiments due to the various materials used, such as mapping populations, experimental designs, statistical methods, and the numbers of traits, markers, and population sizes used in each experiment. The aforementioned localization process frequently requires the genetic analysis of large, isolated populations and a large number of DNA molecular markers, which is time-consuming [16,17].

The method of combining NGS (next-generation sequencing) with mixed grouping analysis, abbreviated as BSA-seq, can quickly and effectively locate genes associated with specific traits. In recent years, this technology has been applied in the study of gene mapping for important agronomic traits in various crops, including Chinese cabbage [18], onion [19], rice [20,21], cucumber [22], and maize [23]. By integrating BSA-seq with RNA-seq, Yan et al. [23] identified *Zm00001d048841* as the most likely candidate gene associated with fasciated ears in maize. Twelve candidate genes related to the early yellowing and senescence of soybean leaves during the late stage of grain filling were found using RNA-seq technology and employing the BSA-seq method for initial localization and map-based cloning [24]. However, so far, there have been no reports on the combined application of the BSA method and RNA-seq when mapping genes for the stay-green traits of maize. In this study, important genes associated with the visual stay-green were investigated; we obtained the F_2_ hybridization between two maize types, stay-green maize (T01) and non-stay-green maize (Xin3). We selected individuals with extreme traits to construct mixed pools for BSA-seq and determined candidate regions. Next, genes related to the visual stay-green were identified via combined RNA-seq analyses.

## 2. Results

### 2.1. Analysis of the Visual Stay-Green

We discovered that two inbred lines exhibited significantly different phenotypes. With respect to T01, a stay-green inbred line, a large portion of leaves remained green in the mature stage, whereas the leaves of the control inbred line (Xin3) were yellow in the mature stage (Figure 1A). The visual stay-green frequency distribution map of 786 populations showed that the average amount of the visual stay-green in the F_2_ population was 51.57%, with a range of 0.00~100.00%. There is an over-parent heterosis in the visual stay-green, and the absolute values of skewness and kurtosis are close to 0 (Table 1), indicating that the visual stay-green conforms to a normal distribution; therefore, we speculate that the main effector gene in this population is responsible for the quantitative trait known as visual stay-green (Figure 1B). The data were then sorted from small to large while 50 plants that stayed green and 50 that did not were chosen to create two extreme pools associated with the visual stay-green trait (Figure 1C).

### 2.2. QTL Mapping Using BSA

To investigate the VSG QTLs, BSA was carried out via bulking 50 NVSG and 50 VSG maize lines from the F_2_ population (Appendix A). To build libraries and resequence DNA, two bulked pools and two parents were utilized. The MGISEQ-2000 platform (MGI Tech Co., Ltd., Shenzhen, China) with high-throughput sequencing yielded an average of 99.85 Gb of data for each library. Based on the read counts, the SNP index and delta-SNP index were calculated for each SNP position. After 1000 permutation tests, we determined the two pools’ ∆ (SNP index) and chose the 95% confidence level as the screening criterion. By combining ED, Ridit, SmoothG, and SNP algorithms, a Manhattan map was drawn to identify the candidate regions associated with these traits (Figure 2). By comprehensively analyzing the correlation regions obtained from the four algorithms mentioned above (Table 2), those related to one candidate interval were finally determined, mainly distributed in 22,466,899~94,938,048 bp of chromosome 9, with a total of 778 genes annotated in the *Zea mays* reference genome of “B73v5”(https://maizegdb.org/, accessed on 14 July 2023).

### 2.3. Transcriptome Sequencing Data Analysis

Twelve cDNA libraries (with two samples and six biological replicates) were sequenced to clarify the molecular mechanisms behind phenotypic variations in visual stay-green. After eliminating any low-quality sequences, clean data with a Q30 ratio of 93.23% were produced. The GC content ranged from 50.49% to 53.68% (Appendix A). Ten DEGs from the RNA-seq data were chosen for the qRT-PCR validation. Figure 3 indicates that the ten selected differentially expressed genes showed consistent trends of upregulation and downregulation in RT-PCR and RNA-seq data. A total of 7796 DEGs were found via differential expression analysis, from which 2425 were upregulated and 5371 were downregulated (Figure 4).

### 2.4. WGCNA of DEGs in Visual Stay-Green

After screening the raw data, 11,378 genes were obtained to conduct WGCNA and were divided into eight modules (Figure 5A). The grey category is an invalid module; the correlation between the genes it contains and one of the important modules is not sufficient. The co-expression network analysis identified eleven modules that were significantly correlated with visual stay-green, with the coefficient varying from −0.86 to 0.69. Modules that are green and brown were significantly positively correlated with visual stay-green, while pink and salmon modules were significantly negatively correlated with visual stay-green (*p* <  0.01, |r| > 0.50, Figure 5B). The green, brown, pink, and salmon modules were composed of 4790 genes, among which the pink module, as the largest module, included 1677 DEGs. All four specific modules were enriched in regulatory pathways related to visual stay-green (Figure 5C), as determined after consulting the relevant literature; the brown module contained four reported stay-green genes, *Zm00001eb103480*, *Zm00001eb135910*, *Zm00001eb142210*, and *Zm00001eb319560*, while the salmon module contained one reported stay-green gene, *Zm00001eb169010*.

### 2.5. Candidate Gene Analysis

To determine the candidate genes associated with visual stay-green, we performed an analysis using BSA-seq, and seven candidate genes that were potentially related to visual stay-green were screened using a functional annotation (Table 3). Through WGCNA, candidate genes related to the visual stay-green were screened using the reported stay-green genes in the tissue-specific module and the highly connected core genes within the module; the brown module used four reported genes to screen out four stay-green candidate genes, while the salmon module used one reported gene to screen out four stay-green candidate genes. The green and pink modules screened one and two candidate genes for the visual stay-green from the high connectivity genes within the module, respectively. Eleven candidate genes related to the visual stay-green were selected from four specific modules (Table 4, Figure 6). To investigate the visual genes influencing maize visual stay-green, we merged the results of BSA-seq and RNA-seq; in these two prospective locations, three genes were examined. Among them, *Zm00001eb378880* mainly expressed the chloroplast, which has the function of regulating the plant’s growth, development, and defense. *Zm00001eb383680* is a gene related to PFPC (phosphoenolpyruvate carboxylase); it can improve photosynthetic capacity and chlorophyll content. *Zm00001eb384100* belongs to the CYP gene family; these are genes related to leaf senescence and which mainly participate in the reaction of jasmonic acid in plants.

## 3. Discussion

### 3.1. Excavation of Key Genes in the Stay-Green Mutant Contributes to the Promotion of Maize

Maize is the largest grain crop planted in China and is an important source of food and industrial raw material. The essence of crop yield formation is the process of the source–sink flow interaction, and the increase in the yield of stay-green crops is directly related to the maintenance of their photosynthetic capacity during the grain-filling period [25,26,27]. Determining the genetic basis of stay-green molecules is beneficial for cultivating new varieties of ideal plant types through molecular breeding methods, thereby comprehensively improving crop resistance to adverse conditions and improving their feeding value and yield potential.

Previously, researchers have attempted to extend the duration of the stay-green mutant to prolong active photosynthesis and cultivate genotypes that can stay green. However, not all stay-green genotypes lead to an increase in grain yield [28,29]. Therefore, there are very few stay-green varieties that can be widely used in production. In this situation, exploring stay-green genes and analyzing the mechanisms of the stay-green mutants using modern methods is conducive to achieving and promoting the high yield of maize.

### 3.2. Integrating BSA Data and RNA-seq Data for Candidate Gene Prediction

The traditional method for identifying QTLs is time-consuming. The method of bulked segregant analysis (BSA), which requires segregating a population, was created to quickly identify the genetic regulation of these features. In recent years, the development of high-throughput sequencing has promoted the application of the BSA-seq method in mapping important traits related to genes in crops. Zhu et al. [30] used BSA-seq to locate 106 candidate regions that modulated salt tolerance in maize seedlings, and by combining transcriptome data, they ultimately classified *Zm00001d053925* as a new functional gene responding to salt stress in the seedling stage. Wang et al. [31] used BSA-seq to locate a candidate region on chromosomes 6 and 9 and, when combined with the transcriptome results, identified *Zm00001d047149*, *Zm00001d046582*, and *Zm00001d047076* as the probable candidate genes for the reversed kernel1 mutant. For gene fine-mapping in maize, combining BSA-seq with RNA-seq is a successful strategy. In this study, our findings suggest that the T01 phenotype is likely caused by three genes, specifically *Zm00001eb378880*, *Zm00001eb383680*, and *Zm00001eb384100*, according to the combination of BSA-seq with RNA-seq.

### 3.3. Chromosome 9 Is Crucial in Controlling Stay-Green-Associated Parameters

A complicated genetic network controls stay-green characteristics [1]. Consequently, determining QTLs is a helpful method for determining its molecular foundation. In recent years, with the use of molecular marker techniques, a plethora of QTL mapping data for the stay-green mutant and its associated features in maize have been discovered. Using RFLP markers, Beavis et al. [32] identified three stay-green QTLs in chromosomes 2, 6, and 9 in an F_4_ population of maize produced from the offspring of the B73 × Mo17 hybrid. Zheng et al. [33] discovered 14 QTLs from the leaf stay-green area on chromosomes 1, 2, 3, 5, 6, 8, and 9 using an F_2_ population from a hybrid between Q319 and Mo17. Wang et al. [12] used 189 F_2_ individuals from A150-3-2 and Mo17 to identify 15 QTLs on chromosomes 1, 4, 5, 6, and 9 for the following variables: the number of green leaves per plant, the total leaf number per plant, and the green leaf area per plant. Yang et al. [14] used an F_3:4_ recombinant inbred line population in the QTL mapping of stay-green traits in maize; they proposed chromosomes 1, 4, 6, 8, and 9 to be important for controlling the stay-green traits of maize. Fang et al. [10] analyzed the formation of QTLs related to the stay-green mutant in maize and found the wx1~umc1107 marker interval on chromosome 9. It is interesting to note that the majority of QTLs discovered in these investigations were found on chromosome 9, suggesting that chromosome 9 is essential for regulating characteristics related to the stay-green mutant. In our study, one major QTL for the traits of leaf visual stay-green was distributed in 22,466,899~94,938,048 bp of chromosome 9; these are precisely the chromosomes of QTL clusters that are closely related to maize stay-green and which have been studied by previous researchers. The above demonstrates the accuracy of QTL localization in this study and confirms the feasibility of the BSA-seq method in quantitative trait localization. The advantage of this study is that it was possible to locate candidate intervals for controlling traits using low-generation materials, which contributes to the limited time taken to obtain the effective QTLs, and the transcriptome data sufficiently demonstrate the accuracy of the data. Due to the limited number of materials available in this study, further verification of the above conclusions is needed. This study could still provide a new approach to maize breeding.

### 3.4. Candidate Gene Analysis

*Zm00001eb378880* is a superoxide dismutase, *Zm00001eb383680* is a phosphoenolpyruvate carboxylase, and *Zm00001eb384100* encodes cytochrome P450 monooxygenase.

*Zm00001eb378880* encodes superoxide dismutase, which is one of the most active antioxidant enzymes involved in the dismutation of oxygen free radicals. Transgenic plants are shown to have a higher stress tolerance when SOD genes are overexpressed [34]. Their tolerance to cold damage and yield were both improved by transgenic Alfalfa plants with about two-fold higher SOD activity than that of non-transgenic plants [35]. The overexpression of MnSOD in tobacco and maize enhances the protection of the plasma membrane and Photosystem II with tolerance to oxygen stress [36,37]. Some studies have also identified that Arabidopsis mutants with strong oxidative resistance generally have a phenotype of the stay-green mutant [38]. We found that the expression level of *Zm00001eb378880* in T01 was higher than that in Xin3; the increase in the expression of this gene could be related to the stay-green mutant.

*Zm00001eb383680* encodes phosphoenolpyruvate carboxylase (PEPC); PEPC is a carboxylating enzyme with important roles in plant metabolism [39,40]. In transgenic lines, the overexpression of SiPEPC led to a two- to six-fold increase in PEPC enzyme activity compared to a non-transformed control. In transformed plants, photosynthetic efficiency was increased and correlated with higher accumulation of chlorophyll, decreased NPQ, increased PSII, and increased ETR [41]. In sorghum, phosphoenolpyruvate carboxylase 3 knockdown has a detrimental effect on growth, productivity, and responses to salt stress [42]. Interestingly, in previous studies, it has been proposed that this gene is a candidate gene for the stay-green mutant [10]; it is necessary to further explore and validate this proposal.

*Zm00001eb384100* encodes cytochrome P450 monooxygenase, which is a B-group cytochrome superfamily protease with heme as a cofactor and which can catalyze various oxidation reactions [43]. The overexpression of the MdCYPM1 gene from apples in Arabidopsis resulted in significant phenotypic changes in transgenic Arabidopsis plants compared to the wild type, which was mainly manifested by a decrease in the leaf chlorophyll content and anthocyanin content [44]. Christ et al. [45] concluded that CYP89A9 is involved in the formation of dioxobilin-type catabolites of chlorophyll in Arabidopsis. We found that the expression level of *Zm00001eb384100* in T01 was lower than that in Xin3. Perhaps, as mentioned above, the low expression level of CYP promotes the accumulation of chlorophyll; accordingly, this gene could play an important role in maintaining green traits.

## 4. Materials and Methods

### 4.1. Plant Materials

The maize inbred line T01 (stay-green) and the control inbred line Xin3 (non-stay-green) were employed in this research. Both have the same genetic background. The stay-green mutant T01 is a stable inbred line that has been continuously self-crossed for multiple generations after heavy ion radiation. The normal phenotype material Xin3 is the sister line of T01. To explore the molecular mechanism of the stay-green mutant in maize, F_2_ segregation progeny, consisting of 786 F_2_ individual plants, was produced for high-throughput sequencing. T01 (stay-green) was crossed with Xin3 (non-stay-green); the former was the female parent and the latter was the male parent. Outstanding F_1_ individuals self-pollinated to produce the F_2_ segregation progeny, which was used for BSA-seq genetics, while plants were grown on the experimental farm of Hunan Agricultural University, China, in 2020.

### 4.2. Statistics of Visual Stay-Green

A total of 786 plants from the F_2_ population and the parents’ visual stay-green were measured after they reached maturity. Every plant in the F_2_ population was measured, and ten plants were chosen by each of the two parents to gauge their visual stay-green. Visual stay-green = the green leaf area during the maturity stage/the green leaf area during the silking stage × 100%; the green leaf area per leaf = leaf length × leaf width × 0.75.

### 4.3. DNA Library Construction and BSA-Seq Analysis

In this study, we numbered F_2_ individual plants during the tasseling stage, extracted DNA from leaves, and selected extreme lines based on their visual stay-green traits at maturity; in the F_2_ population, 50 leaves of extremely stay-green individuals and 50 leaves of non-stay-green individuals were pooled (equal weight) to form the H-pool and L-pools, respectively. Using the CTAB approach, we isolated the parents’ DNA from each sample and created two extreme mixing pools by evenly combining the DNA of green and non-green plants. Using agarose gel electrophoresis and a Nanodrop instrument (Nanodrop Technologies, Wilmington, NC, USA), the purity of the DNA (OD260/OD280) was determined. The DNA was then sequenced on an MGISEQ-2000 platform. After filtering the original reading, the subsequent high-quality clean reading was compared with the reference sequence of B73v5 (https://maizegdb.org/, accessed on 14 July 2023). The GATK software tool (version 3.5-0-g36282e4) was used to detect single-nucleotide polymorphisms (SNPs), which involved marker duplication, local rearrangement, base recalibration variant calls, and SNP filtering. To calculate the SNP frequency (the SNP index), two extreme mixing pools used parents as a reference. In total, 1000 permutation tests were conducted on the calculation results, and a 95% execution level was selected as the threshold to filter candidate regions. Candidate chromosome regions were obtained using the ED, Ridit, SmoothG, and SNP methods.

### 4.4. RNA-Seq Analysis

We collected the ear leaves of plants T01 and Xin3 at 35 days after pollination, with 6 biological replicates for each genotype. The total RNA was extracted from the frozen leaves using an RNA extraction kit (Vazyme Biotech, Changsha, China); the RNA-seq libraries were sequenced using the Illumina HiSeq 2500 Sequencing System (San Diego, CA, USA). Each sample was represented by six biological replicates. The unique reads were then aligned to the maize B73v5 (https://maizegdb.org/, accessed on 14 July 2023) using HISAT2 v2.1.0 with default parameters. Gene expression was measured using fragments per kilobase million reads (FPKM).

### 4.5. WGCNA

A phenotype-weighted co-expression network was constructed in the R Studio package WGCNA with twelve samples and one phenotype, and DEGs with a criterion of FPKM ≤ 1 in each sample were excluded for submission to WGCNA(R, version 4.2.0). Blockwise modules of the automatic network construction function were used to obtain these modules. The correlation r between the ME values of each module and VSG was calculated, with *p* values < 0.05 and |r| > 0.5 considered significant.

### 4.6. qRT-PCR Validation of Genes

Using an RNA extraction kit, the total RNA was isolated from the frozen leaves, and HiScript ^®^ II Q RT SuperMix (Vazyme Biotech, Nanjing, China, R223) was used to synthesize first-strand cDNA in accordance with the manufacturer’s instructions. The actin gene (actin1) served as an internal reference, and 10 specific primer pairs were designed in Primer 5.0 software (Table 5). The 2^−∆∆Ct^ method was used to calculate the relative expression levels of the 10 genes chosen for validation.

### 4.7. Combined Analysis of BSA-Seq and RNA-Seq

The association results of different BSA-seq algorithms in the F_2_ population of maize were compared to screen out the consistent candidate regions and combine functional annotation to screen candidate genes. At the same time, based on the significantly correlated modules obtained using WGCNA (R, version 4.2.0), candidate genes related to visual stay-green were screened using the reported stay-green genes in the tissue-specific modules of maize and the core genes with high connectivity within the modules. We combined BSA-seq and WGCNA to select candidate genes that were consistent.

## 5. Conclusions

In summary, the molecular mechanism underlying maize’s stay-green features was uncovered in this study by using BSA-seq and RNA-seq. The candidate region for stay-green characteristics in maize was found on chromosome 9 based on BSA-seq data. Four modules that had a significant correlation to the visual stay-green were found using RNA data and WGCNA. *Zm00001eb378880*, *Zm00001eb3836800*, and *Zm00001eb384100* were jointly identified by combining BSA-seq and RNA-seq, indicating that these three genes are highly significant for maize greening. This offers a foundation for comparing the genome and transcriptome from the perspective of the genetics of the stay-green mutants.

## Figures and Tables

**Figure 1 ijms-24-17617-f001:**
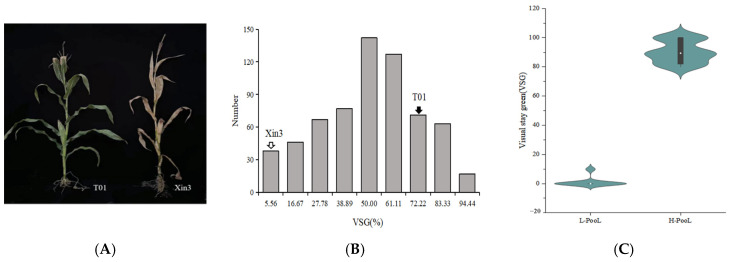
(**A**) The phenotype of visual stay-green trait in T01 and Xin3. (**B**) Frequency distribution of visual stay-green in the F_2_ population of maize. (**C**) The performance of visual stay-green in two extreme mixed pools of visual stay-green.

**Figure 2 ijms-24-17617-f002:**
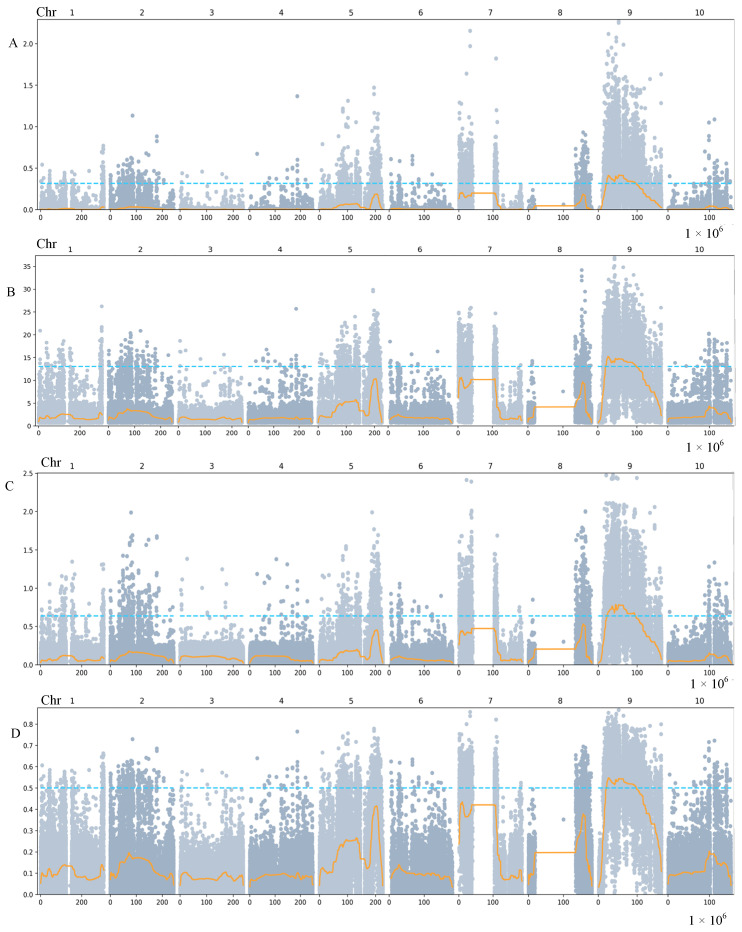
Identification of stay-green gene candidate intervals using four association methods. (**A**) Results of ED correlation. (**B**) Results of Ridit correlation. (**C**) Results of SmoothG correlation. (**D**) Results of SNP correlation. In this graph and the following three, the dots in the figure represent SNP locus, the abscissa is the chromosome position, the vertical coordinate represents the ED or Ridit or SmoothG or SNP value, the yellow line is the fitted ED or Ridit or SmoothG or SNP association value, and the blue dashed line represents the significance association threshold.

**Figure 3 ijms-24-17617-f003:**
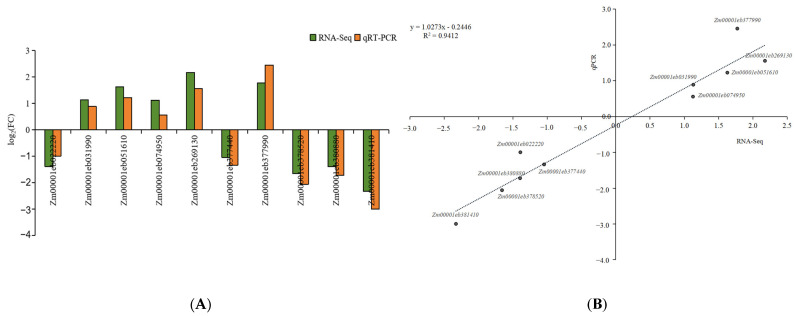
The results of quantitative RT-PCR (qRT-qPCR) validation. (**A**) The RT-qPCR expression of the DEGs. (**B**) Pearson correlation analysis of relative expression levels of genes detected using qPCR and RNA-seq. Note: the dotted line in the (**B**) represent simulated linear equation.

**Figure 4 ijms-24-17617-f004:**
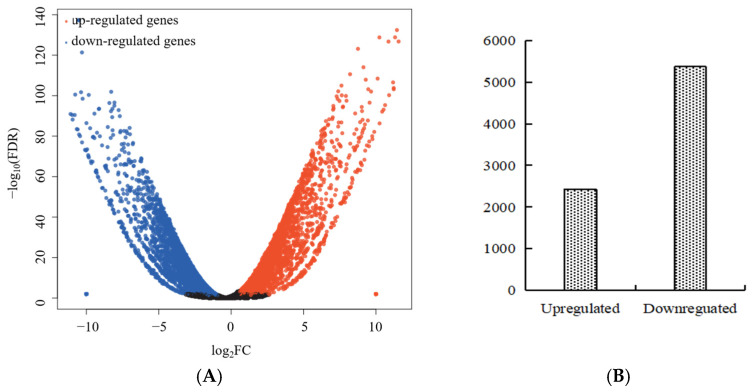
Volcano plot and number of DEGs. (**A**) Volcano plot. (**B**) The number of DEGs.

**Figure 5 ijms-24-17617-f005:**
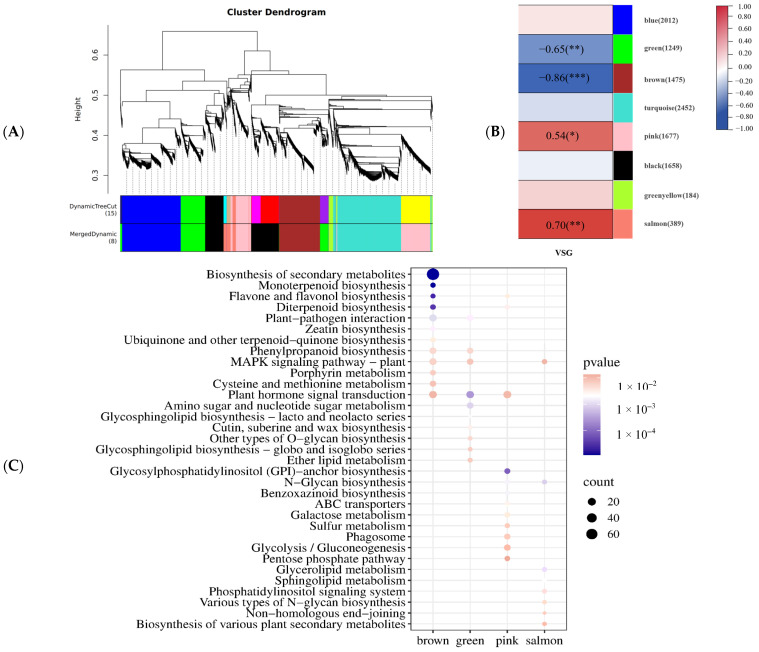
Results of gene co-expression network construction. (**A**) Gene clustering and module construction. (**B**) Correlation between traits and modules (red color of each box represents the positive correlation between module and trait; blue color of each box represents the negative relationships between module and trait; Note: *, ** and ***indicated significant, extremely and highly significant correlation at the 0.05, 0.01 and 0.001 levels, respectively. (**C**) Enrichment pathway of visual stay-green.

**Figure 6 ijms-24-17617-f006:**
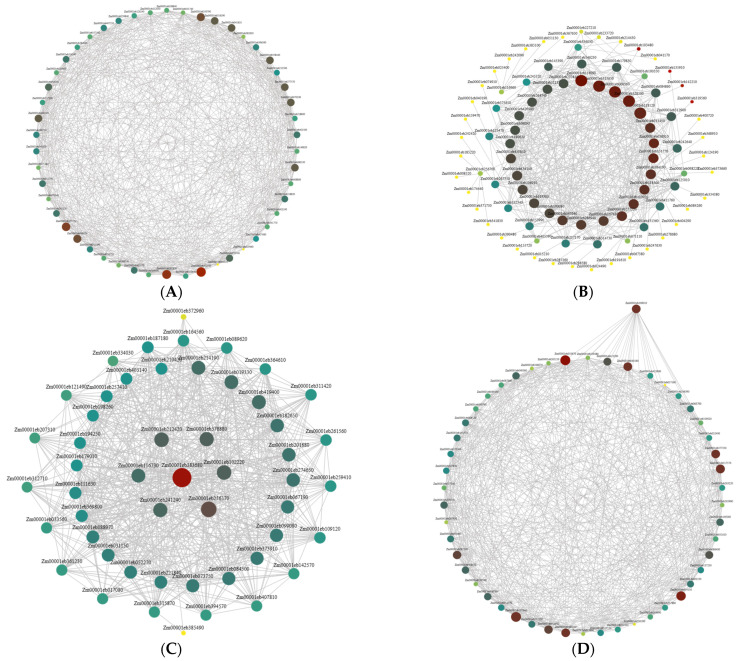
Local regulation network of gene co-expression of key modules. (**A**) Green module; (**B**) brown module, reported stay-green genes: *Zm00001eb103480*, *Zm00001eb135910*, *Zm00001eb319560*, *Zm00001eb142210*; (**C**) pink module; (**D**) salmon module, reported stay-green genes: *Zm00001eb169010*.

**Table 1 ijms-24-17617-t001:** Descriptive statistics of the visual stay-green of F_2_ population and H/L-pool.

Material	Number of Samples	Min (%)	Max (%)	Mean (%)	CV (%)	Kurt	Skew
F_2_	786	0.00	100.00	51.57	48.92	−0.454	−0.374
H-pool	50	80.00	100.00	90.23	8.05	0.200	−1.324
L-pool	50	0.00	10.00	1.18	-	2.418	4.017

**Table 2 ijms-24-17617-t002:** Information of associated regions detected using different methods.

Associated Analysis Method	Chromosome	Start of Associated Regions (bp)	End of Associated Regions (bp)	Gene Number in the Associated Regions
Results of ED correlation	Chr9	21,439,311	97,423,834	841
Results of Ridit correlation	Chr9	20,699,310	98,235,998	895
Results of SmoothG correlation	Chr9	22,466,899	94,938,048	778
Results of SNP correlation	Chr9	20,964,571	98,003,843	878
The intersection of the four association results	Chr9	22,466,899	94,938,048	778

**Table 3 ijms-24-17617-t003:** BSA-seq candidate genes.

Gene	Annotation
*Zm00001eb378880* *	superoxide dismutase
*Zm00001eb380180*	autophagy 2
*Zm00001eb380210*	ribulose 1,5-bisphosphate carboxylase/oxygenase large subunit
*Zm00001eb380300*	senescence-specific cysteine protease SAG12
*Zm00001eb382070*	cryptochrome-2
*Zm00001eb383680* *	phosphoenolpyruvate carboxylase
*Zm00001eb384100* *	cytochrome P450 monooxygenase

* The gene co-mined for BSA-seq and WGCNA.

**Table 4 ijms-24-17617-t004:** WGCNA candidate genes.

Module	Gene	Annotation
green	*Zm00001eb240100*	putative GATA transcription factor 22 isoform X1
brown	*Zm00001eb384100* *	cytochrome P450 monooxygenase
	*Zm00001eb405590*	NAC transcription factor 29
	*Zm00001eb116960*	early light inducible protein1
	*Zm00001eb118120*	agamous-like MADS-box protein AGL8
pink	*Zm00001eb383680* *	phosphoenolpyruvate carboxylase
	*Zm00001eb378880* *	superoxide dismutase
salmon	*Zm00001eb040160*	MYB transcription factor
	*Zm00001eb411260*	diacylglycerol kinase 2
	*Zm00001eb414910*	ZIM-transcription factor 40
	*Zm00001eb332340*	MYB-RELATED transcription factor

* The gene co-mined for BSA-seq and WGCNA.

**Table 5 ijms-24-17617-t005:** Sequences of the specific primer pairs used in RT-qPCR.

ID	Sense Primer	Anti-Sense Primer
*Zm00001eb022220*	GGCCTCAATTCCGTCACTCA	CTTTTTAGCACTGCCGCGTT
*Zm00001eb031990*	AACGGTGTTCTACCCGATGC	TCCATTGGGCTCTTCCTTGG
*Zm00001eb051610*	AGGGATAGTGATGGGCGTCT	TGAGATTTCCTGACATAGCTTTAGA
*Zm00001eb074950*	CGCAGGGGTCTTGAGAGAAG	CTAGCTAGAGCTATCGCGGC
*Zm00001eb269130*	CCGAGAGGGACAGTCCAGTA	GTCGACAAGCTCACGCTTCA
*Zm00001eb377440*	GCGCGCGGGAAATTATAGTG	TGGCCTTCCTCACCTAGCTT
*Zm00001eb377990*	AACCTCGAGAAAGTCTGCGG	CTCAAAGCGATCCCAGGTGT
*Zm00001eb378520*	CGTCAACTGCAACGGGTG	AGCGACGACATGTTCAGAGC
*Zm00001eb380880*	GGCTATGGAAGATGACGCGA	CGACGTCATAAAGTACATAGCGA
*Zm00001eb381410*	TGGTGATGGAGGCTATGGGA	ACAAGGGAGAAGGGTCAACG

## Data Availability

The data presented in this study are available in this article and Appendix A.

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
