# Peer review of "Combined BSA-Seq and RNA-Seq Reveal Genes Associated with the Visual Stay-Green of Maize (Zea mays L.)"

_ijms, 2023, doi:10.3390/ijms242417617_

Round 1
Reviewer 1 Report
Comments and Suggestions for Authors
The idea and the results are very interesting and valuable for scientific audience as well as for maize breeders. Otherwise, the quality of presentation of the research is rather poor. Material and methods are not properly explained. One of the examples: lines 264-268 - It is not clear at all how the samples were taken and prepared for analyses, and it seems that this sentence is not finished. Also, in the Material and methods, section 4.3. DNA Library Construction and BSA-Seq Analysis it is mentioned that 30 individuals of each genotype were chosen for BSA seq, but in the Results section 2.2. QTL mapping using BSA the number of individuals is 50 per genotype. also, it does not seem to be clear how the samples for RNA seq were chosen. It would be good to carefully rewrite the whole Material and methods section, adding more details and explanations in order to be clear how the experiment was performed. In the Results section the results considering quality of sequencing (for BSA seq as well as for RNA seq) are not necessary to be presented. The presentation and explanation of results and discussion part should be improved. Soundness and traceability should be significantly improved. Reference should be checked and corrected. I have noticed some duplicates (for example the 30. and 32. reference are the same). Also, in references 36. and 37. the capital letters should be corrected in small ones.
Comments on the Quality of English LanguageThe quality of English language is very poor and have to be significantly improved. The grammar should be improved and also the terms used for some phenomena must be checked and changed to more suitable. For example, lines 184 and 185 in the manuscript: Determining the genetic basis of stay green molecules... What are stay green molecules? It could be only genetic basis of stay green trait or so. Also, some sentences seem unfinished, I mentioned one of them from the Materials and methods in the Comments and suggestions for Authors secton.
Author Response
Thank you very much for taking the time to review this manuscript.
Comments 1: The quality of presentation of the research is rather poor. Material and methods are not properly explained. One of the examples: lines 264-268 - It is not clear at all how the samples were taken and prepared for analyses, and it seems that this sentence is not finished. Also, in the Material and methods, section 4.3. DNA Library Construction and BSA-Seq Analysisit is mentioned that 30 individuals of each genotype were chosen for BSA seq, but in the Results section2.2. QTL mapping using BSAthe numberof individuals is 50 per genotype. also, it does not seem to be clear how the samples for RNA seq were chosen. It would be good to carefully rewrite the whole Material and methods section, adding more details and explanations in order to be clear how the experiment was performed. In the Results section the results considering quality of sequencing (for BSA seq as well as for RNA seq) are not necessary to be presented. The presentation and explanation of results and discussion part should be improved. Soundness and traceability should be significantly improved. Reference should be checked and corrected. I have noticed some duplicates (for example the 30. and 32. reference are the same). Also, in references 36. and 37. the capital letters should be corrected in small ones.
Response1: Thank you for pointing this out.
My manuscript has been readjusted, especially since the materials and methods you mentioned were not clearly described. Details and explanations have been added, and in the results section, some data presentation has been adjusted. The content of the discussion section has been revised, and the references have been corrected. The specific modifications are presented in red text in the manuscript for your review.
Comments 2: The quality of English language is very poor and have to be significantly improved. The grammar should be improved and also the terms used for some phenomena must be checked and changed to more suitable. For example, lines 184 and 185 in the manuscript:Determining the genetic basis of stay green molecules...What are stay green molecules? It could be only genetic basis of stay green trait or so. Also, some sentences seem unfinished, I mentioned one of them from the Materials and methods in the Comments and suggestions for Authors secton.
Response 2: Thank you for pointing this out. I agree with this comment. Due to the need for moderate or extensive English revisions to my manuscript, I have used the English editing provided by MDPI. Could you please review it again.

Reviewer 2 Report
Comments and Suggestions for Authors
The research is of utmost importance and highlights the molecular basis of visual stay green in maize. However, the manuscript needs major and serious revisions in terms of writing. The sentences are poorly structured and difficult to understand.
The resolution in Figure 6 is very poor.
The highlighted and underlined text need to be seriously improved for clarity and grammar.
Conclusion section needs substantial improvement.

Extensive editing is required to improve the clarity of sentences. Serious need to improve the grammatical errors.
Author Response
Response to Reviewer 2 Comments
|
||
1. Summary |
|
|
Thank you very much for taking the time to review this manuscript. Please find the detailed responses below and the corresponding revisions in the re-submitted files. |
||
2. Point-by-point response to Comments and Suggestions for Authors |
||
Comments 1: The resolution in Figure 6 is very poor. |
||
Response1: Thank you for pointing this out. I have adjusted the clarity of all images in the manuscript, please review. |
||
Comments 2: The highlighted and underlined text need to be seriously improved for clarity and grammar. |
||
Response 2: Thank you for pointing this out. I have carefully considered your highlighted and underlined text and made every effort to correct any unclear descriptions for your review. |
||
Comments 3: Conclusion section needs substantial improvement. |
||
Response 3: I have carefully considered your highlighted and underlined text and made every effort to correct any unclear descriptions for your review. |
||
Comments 4: Extensive editing is required to improve the clarity of sentences. Serious need to improve the grammatical errors. |
||
Response 4: I sought help from MDPI's quick editing service to improve the grammatical. |

Reviewer 3 Report
Comments and Suggestions for Authors
Authors prepared manuscript about maize, their genes related to visual stay green. 45 references were used in the preparation of the manuscript, most of which are really quite new or still relevant.
There is a lot of information in such pictures as figure 6, but the markers are very small and almost impossible to see, maybe it would be possible to arrange them in a different way, so that they could be enlarged at least a little and the text in them could be seen more clearly?
In the discussion, it is difficult to understand where you are talking about the data of this experiment, and where about the results of other, older studies. For example, lines 216-217, are these the data of this experiment? Just insert references to figures, tables. It will be clear where and what data is discussed.
Comments on the Quality of English LanguageEnglish is pretty good, lots of long sentences where it's hard to follow the gist. But the English language does not cause major problems for general understanding.
Author Response
Thank you very much for taking the time to review this manuscript. Please find the detailed responses below and the corresponding revisions in the re-submitted files.
Comments 1: There is a lot of information in such pictures as figure 6, but the markers are very small and almost impossible to see, maybe it would be possible to arrange them in a different way, so that they could be enlarged at least a little and the text in them could be seen more clearly?
Response1: Thank you for pointing this out.
I have adjusted the clarity of all images in the manuscript, please review.
Comments 2: In the discussion, it is difficult to understand where you are talking about the data of this experiment, and where about the results of other, older studies. For example, lines 216-217, are these the data of this experiment? Just insert references to figures, tables. It will be clear where and what data is discussed.
Response 2: Thank you for pointing this out. I have rewritten this section and updated the references. Please review it.

Round 2
Reviewer 1 Report
Comments and Suggestions for Authors
Comments, suggestions and recommendations are accepted and the quality of paper si generally improved and acceptable for publications with minor revisons. There are some ambiguities regarding some parts of the paper which should be clarified.
My comments regarding corrected version of the manuscript follow:
Lines 59-61 – The sentence is not clear: Yan et al.[23] combined BSA-seq-based mapping and RNA-seq profiling to identify causal candidate genes Zm00001d048841 associated with fasciated ears in maize. It should be reformulated.
Line 67 - we obtained the F2 hybridization between two maize types – should be changed in for example: F2 generation has created by crossing two maize genotypes - one stay - green maize (T01) and non-stay-green maize (Xin3).
Comments on Material and methods
4. Materials and Methods
4.1. Plant Materials
The stay-green mutant T01 was mutated due to heavy ion radiation, and the excellent maize inbred line T01 was selected as one of the top ten excellent germplasm resources for crops in Hunan Province, with stay-green leaves in maturity. - It is not clear why the T01 line is mutated? The line chosen for the experimentMo17 is a wild control, with mature leaves appearing yellow in color. –Did you mean that Mo17 was used as wild control in the process of creating T01 mutant? To explore the molecular mechanism of the stay-green mutant in maize, F2 segregation progeny, consisting of F2 individual plants, was produced for high-throughput sequencing. T01 (stay green) was crossed with the Xin3 (non-stay green); the former was the female parent, and the latter was the male parent. Outstanding F1 individuals self-pollinated to produce the F2 segregation progeny, which was used for BSA-seq genetics, while plants were grown on the experimental farm of Hunan Agricultural University in China 2020.
4.4. RNA-Seq Analysis
Ten randomly chosen plants from within each cultivar that had similar development progress were marked in field plots for the trials that followed. Thirty-five days after pollination, they were collected - It is not clear what was collected. Leaves? from ten T01 and ten Mo17 plants, from which six plants of each genotype were chosen for RNA sequencing (RNA-seq). – It is not clear from the previous sentences that genotypes T01 and Mo17 were chosen for RNAseq analysis. The paragraph should start with that statement, and then it should be explained how the samples were chosen for the analysis. The total RNA was extracted from the frozen leaves using an RNA extraction kit (Vazyme Biotech, Changsha, China); the RNA-seq libraries were sequenced using the Illumina HiSeq 2500 Sequencing System. Each sample was represented by six biological replicates. The unique reads were then aligned to the maize B73v5 using HISAT2 v2.1.0 with default parameters. Gene expression was measured using fragments per kilobase million reads (FPKM).
4.6. qRT-PCR validation of genes
Using an RNA extraction kit, the total RNA was isolated from the frozen leaves, and HiScript ® II Q RT SuperMix (Vazyme Biotech, R223) was used to synthesise first-strand cDNA in accordance with the manufacturer's instructions. The actin gene (actin1) served as an internal reference, and ten specific primers primer pairs were designed in Primer 5.0 software (Table 5). The 2–∆∆Ct method was used to calculate the relative expression levels of the 10 genes chosen for validation.
4.7. Combined Analysis of BSA-Seq and RNA-Seq
The association results of different BSA-seq algorithms in the F2 population of maize were compared to screen out the consistent candidate regions; functions were combined to obtain genes and analyze the molecular mechanisms that affect the visual stay green of maize, according to the candidate regions with significantly correlated modules obtained using WGCNA to further determine the genes. – Please, could you explain better how the results of BSA-seq and RNA-Seq were combined? It is not obvious from the previous paragraph.
2. Results
2.1. Analysis of the visual stay green
Line 73 - We discovered that the parents of the visual stay-green… - Please, could you explain what exactly the stated expression means?
Title below Figure 2:
Figure 2. Identification of stay green gene candidate intervals using four association methods. A: 112 ED correlation analysis results, the abscissa is the chromosome position, the ordinate represents the 113 fifth power of the euclidean distance (ED) value after fitting, the yellow line is the fifth power of ED 114 after fitting, the dashed line represents the significance association threshold, the same below. B: 115 Ridit correlation analysis results, C: SmoothG correlation analysis results. D: SNP correlates - Should it be SNP correlation analysis? analysis 116 results.
The same question for Table 2:
It is written for example:
Euclidean distance correlates results, Should it be Euclidean distance correlation results, and so on?
Comments on the Quality of English LanguageMinor editing of English language required.
Author Response
1. Summary |
Thank you very much for taking the time to review this manuscript. |
2. Point-by-point response to Comments and Suggestions for Authors |
Comments 1: Lines 59-61-The sentence is not clear: Yan et al.[23] combined BSA-seq-based mapping and RNA-seq profiling to identify causal candidate genes Zm00001d048841 associated with fasciated ears in maize. It should be reformulated. |
Response 1: Thank you for pointing this out. Revise: By integrating BSA-seq with RNA-seq, Yan et al.[23] identified Zm00001d048841 as the most likely candidate gene associated with fasciated ears in maize. |
Comments 2: Line 67-we obtained the F2 hybridization between two maize types- should be changed in for example: F2 generation has created by crossing two maize genotypes - one stay - green maize (T01) and non-stay-green maize (Xin3). |
Response 2: Agree. Revise: We discovered that two inbred lines exhibited significant phenotypes, with respect to T01, a stay-green inbred lines, a large portion of leaves remained green in mature stage, whereas the leaves of the control inbred lines (Xin3) were yellow in mature stage(Figure 1A). |
Comments 3: It is not clear why the T01 line is mutated? |
Response 3: Thank you for pointing this out. I found an issue with the description here. Revise: The maize inbred line T01(stay green) and the control inbred line Xin3 (non-stay green) were employed in the research. Both have the same genetic background. The stay green mutant T01 is a stable inbred line that has been continuously self crossed for multiple generations after heavy ion radiation. The normal phenotype material Xin3 is the sister line of T01. |
Comments 4: It is not clear from the previous sentences that genotypes T01 and Mo17 were chosen for RNAseq analysis. The paragraph should start with that statement, and then it should be explained how the samples were chosen for the analysis. |
Response 4: Thank you for pointing this out. Revise: Collect the ear leaves of plants T01 and Xin3 at 35 days after pollination, with 6 biological replicates set for each genotype. |
Comments 5: Please, could you explain better how the results of BSA-seq and RNA-Seq were combined? It is not obvious from the previous paragraph. |
Response 5: Thank you for pointing this out. Revise: The association results of different BSA-seq algorithms in the F2 population of maize were compared to screen out the consistent candidate regions, and combine functional annotation to screen candidate genes. At the same time, based on the significantly correlated modules obtained using WGCNA, candidate genes related to visual stay green were screened using the reported stay green genes in the tissue specific modules of maize and the core genes with high connectivity within the modules. Combined BSA seq and WGCNA analysis to select candidate genes that are consistent. |
Comments 6: Title below Figure 2, SNP correlates- Should it be SNP correlation analysis? |
Response 6: Modified |

Reviewer 2 Report
Comments and Suggestions for Authors
Can be accepted in its present form.
Author Response
Thank you again for your review